# Experiences of Older Adults with Frailty Not Completing an Advance Directive: A Qualitative Study of ACP Conversations

**DOI:** 10.3390/ijerph19095358

**Published:** 2022-04-28

**Authors:** Zoe Wan, Helen Y. L. Chan, Patrick K. C. Chiu, Raymond S. K. Lo, Hui-Lin Cheng, Doris Y. P. Leung

**Affiliations:** 1School of Nursing, The Hong Kong Polytechnic University, Hong Kong 999077, China; zoe-sf.wan@polyu.edu.hk (Z.W.); eileen.cheng@polyu.edu.hk (H.-L.C.); 2Nethersole School of Nursing, The Chinese University of Hong Kong, Hong Kong 999077, China; helencyl@cuhk.edu.hk; 3Li Ka Shing Faculty of Medicine, The University of Hong Kong, Hong Kong 999077, China; chiukc@ha.org.hk; 4Department of Hospice and Palliative Medicine, Shatin Hospital and Bradbury Hospice, Hong Kong 999077, China; losk@ha.org.hk

**Keywords:** geriatrics, frailty, advance care planning, end-of-life care, qualitative research, thematic analysis

## Abstract

Advance care planning (ACP) facilitates individuals to proactively make decisions on their end-of-life care when they are mentally competent. It is highly relevant to older adults with frailty because they are more vulnerable to cognitive impairment, disabilities, and death. Despite devoting effort to promoting ACP among them, ACP and advance directive completion rates remain low. This study aims to explore the experiences among frail older adults who did not complete an advance directive after an ACP conversation. We conducted a thematic analysis of audiotaped nurse-facilitated ACP conversations with frail older adults and their family members. We purposively selected ACP conversations from 22 frail older adults in the intervention group from a randomized controlled trial in Hong Kong who had ACP conversation with a nurse, but did not complete an advance directive upon completing the intervention. Three themes were identified: “Refraining from discussing end-of-life care”, “Remaining in the here and now”, and “Relinquishing responsibility over end-of-life care decision-making”. Participation in ACP conversations among frail older adults and their family members might improve if current care plans are integrated so as to increase patients’ motivation and support are provided to family members in their role as surrogate decision-makers.

## 1. Introduction

Frailty is an age-associated condition characterized by a decrease in biological reserves, resulting in unstable homeostatic mechanisms [1]. Hence, older people are more vulnerable to abrupt declines in health, cognitive impairment, disabilities, and death [2,3]. Frailty is a common geriatric syndrome that affects approximately 26.8% of older people globally [4].

Advance care planning (ACP) is a process of communication aimed at helping individuals to proactively make decisions on their end-of-life (EOL) care when they are mentally competent. The process enables individuals to clarify their wishes, values, and beliefs; therefore, their participation in the process is essential. Given the vulnerability of older people with frailty, ACP is particularly relevant to this population, as it enables individuals to define EOL care, clarify their values with family and healthcare providers, and to record these preferences in their medical records or in an advance directive [5]. It highlights the concept of person-centered care in emphasizing that EOL care should align with personal care wishes and preferences [6,7]. There is growing evidence demonstrating the benefits of ACP in terms of increasing documentation of EOL care decisions and improving patient–surrogate congruence in relevant decision-making [8,9,10,11].

Despite these merits, there is a great deal of diversity among older people and patients with advanced illnesses in their level of readiness for ACP [12,13,14]. While a considerable proportion of this population appreciated having the opportunity to plan ahead for their future care, many were resistant or remained in the stage of contemplation [15]. The barriers that were identified mainly related to inadequate knowledge about ACP, disease trajectories, and prognosis and treatment options [16,17,18]. However, studies showed that relevant educational interventions have limited effects on increasing participation in ACP or on improving care outcomes [19,20].

Another barrier commonly identified as a hindrance to ACP was insufficient professional services or family support [18,21]. Many healthcare providers were reluctant to broach the topic of ACP due to time constraints and to a perception of their lack of competence to properly deal with the issue, whereas family members worried that the process would be distressing [16]. A number of initiatives for providing ACP training to healthcare providers and specific personnel to facilitate family communication were launched to develop an environment conducive to discussing ACP. ACP or advance directive (AD) completion rates, however, remain low in regions with relevant legislation and available services [22,23]. Some regions such as China including Hong Kong have neither statutes nor case laws on the legal status of AD, which resulted in a majority of citizens who had not heard about AD or did not prefer to document their wishes in advance, although many of them indicated they would willing to discuss their future health care plan when opportunities were offered after informed of ACP/AD [16,24]. In Hong Kong, the AD documentation rate was very low, with only 3,275 AD documents recorded in 30 hospitals in 2017 [25].

Much attention and effort has been devoted to promoting ACP over the last decade, but progress seems to have stalled. The lessons learnt, particularly from unsuccessful experiences in ACP discussions, might provide insights on what does not and the underlying reasons. Hence, in this study, we attempted to use a qualitative approach to understand the failure of ACP among participants who were regarded as unsuccessful cases. We defined unsuccessful cases as those who were not interested in documenting their EOL care preferences after a structured ACP program. Here, we present the findings from qualitative analysis of the audiotaped and transcribed content of the face-to-face ACP conservations of those unsuccessful cases from a randomized controlled trial (RCT) evaluating the effects of a nurse-led video-supported ACP program on older adults with frailty, with completion of an advance directive as the primary outcome [26].

## 2. Materials and Methods

### 2.1. Design

We conducted a thematic analysis of audio-taped nurse-facilitated ACP conversations with frail older adults, and their family members in some cases, obtained through an RCT (Chinese Clinical Trial Registry, Ref No.: ChiCTR-IOR-17012341 on 11 August 2017). The original RCT study started subject recruitment in a medical ward of a public hospital in Hong Kong since December 2018. The target sample size was revised from 298 to 148 with the approval from the project funder on 11 July 2019. However, the subject recruitment has been suspended since January 2020 due to the outbreak of COVD-19 pandemic, leading to the cessation of all research activities in the study hospital and home-based intervention delivery in the RCT. The original RCT is expected to be terminated prematurely by May 2022 due to the end of the funding. During this period, a total of 105 older adults with frailty were enrolled into the RCT, 51 were allocated to the experimental group with 33 had received the ACP intervention, 20 refused to receive the intervention (most because of the social event in 2019 in Hong Kong), 1 passed away and 5 were pending for intervention due to the COVID-19 situation. Among the 33 participants that had received the ACP intervention in the experimental group, 29 allowed us to audio-record the conversations. We purposively selected recordings of those who had held an ACP conversation with a nurse facilitator of the research project at least once, but did not achieve the primary outcome of the RCT upon completing the intervention (i.e., unsuccessful cases), in that they decided not to complete an advance directive. The dialogues among the participants, their family members, and the nurse facilitator during the discussion were analyzed to explore the challenges of conducting an ACP conversation in particular the experiences of frail older adults who did not complete an AD upon the conversation. Results of the trial are reported in a separate article, and the current analysis was not included in the plan of the RCT.

### 2.2. Participants

Participants were eligible for the original RCT if they were in-patients aged 60 or above, fulfilled at least one criteria of the FRAIL scale [27], were clinically stable, able to communicate in Chinese, and cognitively intact (MMSE > 17) [28]. Those who had already signed an advance directive or who had been referred to a palliative care service during the study period were excluded. They could invite family members to join the ACP conversations with the nurse facilitator. The criteria for the inclusion of family members were those aged 18 years or above, able to communicate in Chinese, and who had been invited by the patient to join the conversation. In this qualitative exploration, we selected the dialogues of participants in the experimental group who did not complete an AD after having the nurse-led ACP conversations for analysis.

### 2.3. ACP Discussion Intervention

Frail older adults in the experimental group of the RCT were scheduled to have two 1 h face-to-face sessions of ACP conversations at home, one week apart, after being discharged from the hospital. The ACP sessions were facilitated by a trained nurse facilitator following a standardized protocol. The nurse facilitators had more than ten years of clinical experience and had attended a two-day training workshop on ACP prepared by our research team to equip them with the knowledge and skills necessary to hold an ACP discussion. The ACP intervention consisted of four topics: (1) view on current health status, (2) current wishes and values, (3) view on EOL treatments, and (4) view on advance directives, supplemented with a 5 min video showing EOL treatment options. There were no restrictions on the order in which the four topics were to be discussed in the intervention. A maximum of two sessions were offered as part of the trial design, and an information booklet on these four topics was available for participants receiving the intervention. During the ACP conversation, older adults were encouraged to write down their values, wishes, goal of care and future healthcare plan, if any, in the booklet. For those who wished to document their future healthcare decisions, a structured AD form adapted, with permission, from the modified directive model form by Hospital Authority of Hong Kong was used. In the structured AD form, options for refusing of the use of life-sustaining treatment or continuing to receive basic and palliative care under three cases of terminally ill, persistent vegetative state of irreversible coma, and other end-stage irreversible life-limiting condition, separately. The ACP discussion intervention is specially designed for the older adults, with no component targeting family member because the trial only recruited older adults with frailty without a family member. However, the presence of family members in the ACP discussion intervention was encouraged because the older adult’s value, beliefs, and EOL decision could be known to the family member who is likely to be the surrogate for decision making at a later stage of life of the older adult, thereby reducing the anxiety of the family member (for addressing a secondary outcome of the original trial). Details of the protocol have been presented elsewhere [24]. Before the start of each ACP conversation session, oral consent to record the whole conversation was obtained from the patient and their family member(s), if any. Only when such consent was obtained from all of the participants in the ACP session was the session audio-recorded. The average length of the face-to-face session was 41 min, ranging from between 16 and 70 min.

### 2.4. Data Analysis

We adopted thematic analysis by Braun and Clarke [29] to analyze the transcripts of ACP conservations. First, three members of the research team (ZW, DL, HC) familiarize with the data by listening to the recordings and read transcripts several times to ensure that we fully understood the content and to aid in immersion. We generated an initial list of codes based on the first few transcripts independently and then had multiple meetings and email conversations to discuss these initial codes and interpretation. Second, we coded the following transcripts using the preliminary coding framework and condensed codes with similar meaning into subthemes and themes. The coding process was iterative. The team continuously compared and discussed the coded sentences and refined the subthemes and themes until they reached an agreement. Quotations were selected through a critical discussion among the authors. The quotations were translated into English by the first author, confirmed by the other two members to ensure semantic equivalence and finally proofread by all co-authors.

### 2.5. Trustworthiness

Measures suggested by Shenton [30] were implemented to ensure the trustworthiness of this study. Credibility was achieved by investigator triangulation, as three investigators participated in coding to avoid idiosyncratic interpretations and biased decisions [31]. Before commencing the ACP conversation, individuals were given the opportunity to refuse to participate. Those who elected to join the conversation indicated that they were willing to express their true thoughts and feelings. This ensured the honesty of the participants in providing data. Transferability was achieved, with clear and consistent patterns emerging from the data. Dependability was achieved with detailed information about the participants, the research design, and the data collection all being well documented to ensure that the process was logical and traceable. The transcripts of the ACP conversations with the participants were reviewed by three independent researchers. The researchers held multiple discussions to minimize biases in coding. There was a high level of consensus on the independent coding between the coding team and the whole team, ensuring confirmability of the interpretations.

### 2.6. Ethical Considerations

Ethical approval for the study was obtained from the university and the research ethics committee of the participating hospital. Written informed consent was obtained from all of the participants and family members who joined the ACP intervention. All personal identifiers were removed before analysis to ensure privacy.

## 3. Results

### 3.1. Participants

Among the 29 participants in the experimental group who agreed to have their ACP process recorded, 22 (75.9%) decided not to complete an advance directive upon completing the ACP intervention. Table 1 shows their socio-demographic and clinical characteristics and participation information in the ACP conversations. These participants were aged between 67 and 91 years, with an average of 79.3 years (SD 6.6). Approximately two thirds of them were male (68.2%) and most of them were married (81.8%). Many of them had a secondary education or above (40.9%) and did not have any religious beliefs (45.5%). The majority (86.4%) were living with family members or significant others. Most were living with multi-morbidities (i.e., two or more chronic conditions, mainly heart disease, diabetes, or hypertension). Their mean FRAIL scores were 2.36 ± 1.09 out of 5 ranging from 1 to 4.

Among these 22 participants, 8 (36.4%) completed the ACP in a single session and the remainder in two sessions. The second session was needed mostly because (1) the participants needed more clarification about the concept of ACP and (2) the participants wanted to invite family members who had not attended the first session to join the ACP conversation. Fifteen participants (68.2%) attended the ACP intervention in the presence of members of their family, who were their spouse, children, and/or domestic helper. In general, the level of participation of the family members was low, with family members of two frail older adults remaining almost silent throughout the ACP conversations.

### 3.2. Themes

Three themes concerning reasons for the failure to complete an advanced directive after the ACP discussions were identified from the transcripts of the conversations during the intervention: (1) refraining from discussing EOL care, (2) remaining in the here and now, and (3) relinquishing responsibility over EOL care decision-making. The subthemes within each theme are shown in Table 2.

#### 3.2.1. Theme 1: Refraining from Discussing EOL Care

The participants expressed unrealistic hope for their health and limitations of individual competence in medical decision-making that hindered their engagement in discussing options for their EOL care and treatments.

##### Wishful Thinking

During the ACP conversations, most of the older adults generally hoped to have better health and to maintain a good level of activity in daily life. Such an optimistic view about their own health prevented them from discussing EOL care. When talking about their future health, many participants remained optimistic. They constantly expressed a wish to live well. For example, an 84-year-old female participant said:


*“I want my health to be better and better, without pain … [I want] to maintain my health condition … no more worsening…. I would like to take care of myself.”*
(P3)

##### Feeling Uncomfortable with the Topic

The majority of participants felt uncomfortable with the topic of EOL care and did not want to discuss it openly. A few participants felt that it was unnecessary to discuss EOL care options because they thought that ACP would interfere with their intended goal of treatment and care, and they wished to extend their life at all costs. In some cases, both patients and family members worried that having an ACP conversation would lead to misfortune or affect health. A 70-year-old male participant said:


*“I am old with ill-health. I have a headache when I think of this topic.”*
(P15)

A daughter rejected the idea of showing the video about EOL care to her father because it might affect his recovery. She said,


*“He can’t understand what you have just said. He would not be able to imagine what would happen by then. He didn’t think about such poor situation [health status]…. He cannot digest the information you told, I think. He is not interested in these, probably he is not in a good mood. I don’t want him to think too much. The more he think, the more difficulty for him to get recover. The video may make him more depressed.”*
(C17)

##### Finding the Information Difficult to Understand

Some participants said that they lacked knowledge relating to ACP, and thus were unable to make EOL care decisions. Both older adults and family members felt that some related concepts for example, advance directives and EOL care, were difficult to understand. A 73-year-old lady noted:


*“It isn’t necessary to think about EOL care. First, I don’t know how to choose [among the treatment and care options]. Second, I don’t understand. I will think about it when the time comes. I don’t have plans for that yet.”*
(P21)

A son shared his thoughts about the challenges of having ACP with his father:


*“It is difficult for him to understand the concept of EOL care and advance directives. It is also difficult for us to these to him.”*
(C2)

##### Viewing Saving Life as an Overriding Goal of Care

Some participants believed that saving lives should be the overriding goal of medical care. They regarded life as precious and felt that every means should be attempted to prolong it, regardless of the consequences of the treatments. For example, an 89-year-old man was surprised to learn about the idea of forgoing life-sustaining treatment. He exclaimed,


*“So you teach me to refuse resuscitation? [Then,] how can the doctor save you [my life]?”*
(P10)

#### 3.2.2. Theme 2: Remaining in the Here and Now

The participants remained at the moment as they felt they were still healthy. Instead of thinking about the future, which is out of their control, they were inclined to focus their attention on the present moment. Hence, they considered ACP unnecessary for now.

##### No Urgency in Discussing ACP

Many participants said that they found it difficult to predict the future, especially given the condition of their health. They were satisfied with their current health status and did not see the urgency of having an ACP conversation regarding their EOL care preferences. A 78-year-old female participant said,


*“It’s hard to tell [the future] as my health condition could change.… It is difficult to tell you my EOL care preference. I have not thought about it because I am still healthy. I don’t know how to set EOL care goals.”*
(P4)

##### Letting Nature Take Its Course

They also tended to use the phrase “let it be” to express their thoughts about accepting the natural process of becoming sick and dying as they age. After watching the video about EOL care treatment options, an 80-year-old lady stated,


*“I don’t have any opinions on EOL care. Just let it be! I will think about it when my health gets worse…. God has a plan on when to take you away. When it is the time for you to leave the world, you’ll have to.”*
(P19)

##### Living in the Present Moment

Many participants were more interested in thinking about how to maintain their health and quality of life as good as possible at the present stage and did not want to bother about the future. A 75-year-old lady described the importance of good health:


*“Health is the most precious thing…. The most important thing for me is that I can take care of myself and dine in a restaurant. Other than that, I have nothing to hope for.”*
(P7)

Another 81-year-old man, concerned about what matters most to him at the present moment, said:


*“Regarding my health, it is difficult to tell about the future … There is no doubt that being able to eat, sleep, and walk are the most important and essential things for living now.”*
(P22)

#### 3.2.3. Theme 3: Relinquishing Responsibility over EOL Care Decision-Making

Some participants relinquished their right to plan for EOL care and intended to designate the responsibility to others, usually medical doctors or family members, to make in-the-moment medical decision.

##### Having Trust in Healthcare Professionals

Instead of indicating their own preferences, the participants indicated that they respected the medical profession and trusted that their doctors would make the “right” decision for them. An 89-year-old man stated that:


*“I won’t make a decision on EOL care beforehand. Let the doctor decide [for me]. I consult the doctor because I trust him. I will agree with his decision.”*
(P10)

##### Allowing Flexibility for Family Members

Some participants worried about becoming a burden on their family when they become seriously ill. Instead of making decisions in advance, they believed that letting their family members be the surrogate decision-makers would give them more leeway in the decision-making process. For example, an 81-year-old man said:


*“My views … by the time … let my son make the decision [for me] …. At the last stage of life, suffering would only last for a short period of time. If you keep on like this [being placed on life-sustaining treatments] … my children would be heartbroken when seeing that situation, so let them decide which treatment would be the best [for me] by then. Not necessary to think much about it [now].”*
(P12)

##### Supporting Shared Decision-Making

A few participants supported shared decision-making between the medical doctors and their family members. They believed that doctors together with their family members could make a decision on EOL care for them. An 86-year-old man said:


*“I will agree with the doctors’ decisions. My son and my wife can decide on EOL care for me too…. I will let my wife to decide because I am not capable of making decisions.”*
(P6)

## 4. Discussion

In this paper, we reported the themes identified in the conversations during a structured video-supported ACP intervention among Chinese older adults with frailty who decided not to complete an advance directive in a RCT [26]. Through thematic analysis, our findings showed that the three key themes that emerged throughout the conversations were refraining from discussing EOL care, remaining in the here and now, and relinquishing responsibility over EOL care decision-making.

Although we have successfully initiated ACP conversations with older adults with frailty who had been hospitalized, a large proportion of them finally turned into unsuccessful cases. Instead of feeling anxious about death, the participants generally accepted that getting old and becoming frailer is a natural process. However, they did not see the clinical relevance of ACP for themselves, either because they believed that they were still healthy and that it was too early for them to think about the future or because they wanted to focus on their current needs in daily livings. Such an observation is in line with the findings from previous studies targeting older adults with frailty in the community [32,33]. ACP is still a new concept to most Chinese [34,35]. These findings support the inclusion in an ACP discussion of not only a future care plan, but also of the current care plan to create relevance to motivate older adults with frailty to participate in the conversations [33]. Integrating topics in the current care plan about the risk of one’s health deteriorating and of becoming mentally incompetent can increase the sense of relevance in the topic of ACP [1,36].

Many of our participants had relinquished responsibility over their EOL care decision-making, although the ACP intervention provided them with an opportunity to exercise autonomy in planning their EOL care. They preferred to allow their doctor or their family or the two parties together to make the decision for them. The study by Cheung et al. also reported the limited participation that Chinese palliative care patients had in making autonomous decisions [17]. These findings support the importance of family involvement in ACP conservations with older adults with frailty. Recent studies have consistently shown that individuals in both Western and non-Western populations want to consider the opinions of family members to support them in making decisions regarding EOL care and treatment options [22,37]. The ACP models should be expanded from advocating individual self-determination to relational autonomy, which supports communication among individuals, family members, and healthcare [38,39]. A local study of geriatric inpatients reported that “family will decide for me” was the main reason for declining to engage in ACP [40]. In line with a recent study in China [41], the participation in the ACP conversations among the family members in the analysis was low. One of the family members in our analysis indicated that it was a challenge to explain ACP-related concepts to their older relatives. Previous studies also reported that many family members were unprepared to discuss ACP with their older relatives, for the reasons could include the difficulty of thinking about death and dying and believing that ACP is irrelevant [42]. They need to clearly understand a patient’s values and beliefs regarding care preferences, and the rationale for making decisions on EOL wishes, in order to prepare themselves to act as substitute decision-makers. Enhancing health literacy in family members for EOL care is thus very important, as the discussion will inevitably involve medical knowledge and the use of technical terms when describing the EOL situation and the related treatment options. Given longer life expectancies, many adult children will sooner or later become family caregivers who will be called upon to support older relatives. More studies on supporting family members in their role as surrogate decision-makers for their older relatives are needed [13]. Another priority should be providing community education to young adults to raise their awareness of the importance of engaging in ACP in the capacity of a family member in EOL care [43]. Low public engagement in ACP is not limited to regions without legislation establishment for ACP/AD. A recent study in Ireland reported that many adults, despite recognizing ACP as important, were reluctant to broach ACP because it linked to EOL care, funeral plans, and would cause distress to their loved one [15]. Thus, providing educational opportunities for community organizations could be a way to raise awareness and encourage public engagement in ACP [44,45]. A community action approach to promote early ACP conversation through public education by shifting ACP from a health issue to a ‘normal’ conversion is advocated [15].

This study is the first to explore the views of older adults with frailty on ACP based on their actual dialogues during the ACP process. The audio-recordings of the ACP conversations provide a great deal of detail on how such older adults responded in a real encounter with the topic of ACP. The participants had a wide range of disease types and frailty levels, which led to variations in the topics of concern to the participants. However, this study had several limitations. Our qualitative findings should be viewed in light of a self-selection bias in that the patients and family members in this study may have been better prepared on this topic than others because of the informed consent process for participating in this study. In addition, the structure of the AD form might have impacted the older adults’ willingness for completion because of the limited available options for future treatment plans for selection. More specifically, the option of using life-saving/sustaining treatment was not included, resulting in excluding older adults with life-saving as their goal of care from the current analysis. Subjects were recruited from one medical ward in Hong Kong. Further research is required to assess whether the findings can be applied to other care settings or other Chinese communities. In addition, the majority of the participants were male and married. It remains unclear if the responses would be influenced by gender or the availability of family support. More female and single participants should be involved in a future study.

## 5. Conclusions

This paper describes experiences of failure when engaging in ACP conversations among older adults with frailty in Hong Kong. These findings improve our understanding of the reasons hindering the completion of an advance directive during ACP conversations. To encourage more participation in discussing preferences in EOL care and treatment in ACP conversations for this target population, we recommend integrating current care plans in ACP conversations to create relevance for patients and to provide support to family members in their role as surrogate decision-makers.

Several initiatives could be used to promote engagement with ACP in older adults with frailty. To integrate current care plan for older adults with frailty, raising the capacity among healthcare professionals for ACP facilitation is the key. Treatments to some common complains in older adults such as pain, sleep difficulties could be highlighted in training for ACP facilitation. Family members could play a key role in influencing attitudes and engagement with ACP. Educational seminars or workshops introducing the concept of ACP and what it means to young adults can raise their own awareness of ACP as well as equip them with the knowledge so that they can help their older relatives to review some of the core elements of ACP including values, beliefs, and what can contribute quality of life in normal conversations by not touching EOL treatment options. Initiating ACP in a non-threatening way by their loved ones may also reduce older adults’ reluctance towards engagement. At the policy level, establishing the legal status of AD and providing more EOL care treatment options in the AD document is required.

## Figures and Tables

**Table 1 ijerph-19-05358-t001:** Socio-demographic, clinical characteristics and participation in ACP conversation of the participants (N = 22).

Participant	Age	Gender	Marital Status	Education Level	Living Alone	Religion	No. of Diagnosis	Frailty Score	No. of ACP Session	Family Member Present at the ACP
1	68	M	Married	<Primary	No	Buddhism	5	2	2	Wife
2	87	M	Married	Secondary	No	No	5	3	2	Son
3	84	F	Widow	<Primary	No	Christianity	8	4	2	2 daughters & a son
4	78	F	Married	<Primary	No	Buddhism	3	1	2	Husband
5	72	M	Married	Secondary	No	Buddhism	5	2	1	Wife
6	86	M	Married	<Primary	No	No	4	2	1	Wife and son
7	75	F	Widow	Primary	Yes	No	2	3	2	No
8	77	M	Married	Primary	No	Christianity	9	3	2	Wife
9	78	M	Married	Primary	No	Christianity	5	1	2	Wife, daughter & domestic helper
10	89	M	Married	Primary	No	No	3	2	2	Wife
11	84	F	Widow	>Secondary	Yes	No	5	2	1	No
12	81	M	Married	Primary	No	Ancestor worship	3	3	2	Wife
13	77	M	Married	Secondary	No	Catholicism	11	2	2	Wife & son
14	79	F	Married	>Secondary	No	Catholicism	4	3	2	No
15	70	M	Married	<Primary	No	No	7	3	2	Wife
16	84	M	Married	Secondary	No	No	7	4	1	No
17	91	M	Married	Secondary	No	No	8	1	1	Wife & daughter
18	83	M	Widower	<Primary	Yes	Ancestor worship	8	1	1	No
19	80	F	Married	>Secondary	No	No	3	1	1	Husband
20	67	M	Married	>Secondary	No	Catholicism	2	1	2	No
21	73	F	Married	<Primary	No	No	4	4	2	Husband & domestic helper
22	81	M	Married	Primary	No	Ancestor worship	5	4	1	Wife

**Table 2 ijerph-19-05358-t002:** Themes and subthemes.

Theme	Subtheme
Refraining from discussing EOL care	Wishful thinking
	Feeling uncomfortable with the topic
	Finding the information difficult to understand
	Viewing saving life as an overriding goal of care
Remaining in the here and now	No urgency in discussing ACP
	Letting nature take its course
	Living in the present moment
Relinquishing responsibility over EOL care decision-making	Having trust in healthcare professionals
	Allowing flexibility for family members
	Supporting shared decision-making

## Data Availability

The data underlying the results presented in the study available from the principal investigator subject to approval from HKU/HA HKW IRB (Rm 901, Administration Block, Queen Mary Hospital, Pokfulam, Hong Kong), Joint CUHK-NTEC CREC (8/F, Lui Che Woo Clinical Sciences Building, Prince of Wales Hospital, Shatin, Hong Kong) and HSEARS (c/o Research Office, The Hong Kong Polytechnic University, Hunghom, Hong Kong), which gave permission for the study.

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
