# Peer review of "Experiences of Older Adults with Frailty Not Completing an Advance Directive: A Qualitative Study of ACP Conversations"

_ijerph, 2022, doi:10.3390/ijerph19095358_

Round 1
Reviewer 1 Report
This is a well-written report on a study on ACP in persons with frailty. To address the relevant research question of why so few persons were interested in writing up advance plans even when they had ACP conversations, it makes sense to purposefully select these persons for qualitative interviews. Fortunately, the participants agreed to another interview which allowed for collecting these valuable data. The methods are thorough, reference is made to a protocol article in JAN and ethics review.
I have one main point and a few minor suggestions to improve the manuscript.
- The main point is that the contents of the intervention does not line up with the responses of the persons with frailty. They perceive written plans to be advance directives about particular medical interventions at the end of life, probably triggered by the topics of the videos shown to the participants. However, in the protocol article, component 2 was “Clarify patient's personal values and beliefs underpinning care preferences.” From the quotes, we can see that persons did not refer to writing up possible goals of care, documenting values, or even indicate which family member should make decisions for them. This raises questions as to whether the ACP intervention was in fact focused on limiting life-sustaining interventions rather than a conversation about what is important to people, what makes quality of life and how could care and treatment support this in second instance. Indeed, the subtheme “viewing life saving as an overriding goal of care”, if so sure about this, why did people not want to write up this one preference? Was it because it did not occur as an option in the encounter, this not being part of a standard form, so participants were not offered this option? Or if completing a standard form, did they also have to have an opinion about the specific treatments and limiting it to writing up their main goal was not possible or they may have felt this was inappropriate as incomplete? Same with “Allowing flexibility for family members.” Please clarify the focus of the intervention, was it the same as the main outcome (just completion of an advance directive; with multiple options that needed to be ticked off in a pre-structured format?). If participants really perceived this as a purely medical issue, and a pre-structured format was directive to what needed to be completed, please refer to this in the limitations section.
Minor
- The title may be appealing and popular or even blaming the participants but does not cover the contents of “reality” at that point because the ACP conversations are not about current care, but about future care, even possibly more distant end-of-life care.
- Please report the overall response rate to show readers to what extent the participants of the RCT may have been a selected group in itself.
- Table 1. For a denominator of 22 persons, providing decimal percentages and numbers suggests an inappropriate level of precision. Please round off.
- Language of citations is generally understandable in English as translated by the researchers themselves. Some impress doubtful to me (for example, “won’t think about his health status in a bad way…. “; do you mean he did not express himself in a negative way or something like that or was it expressed in childish language and you are trying to convey that?). Please have a professional translator check the quotes as it is so important to have these right.
- Family mostly remained silent. They were not being addressed by the facilitator at all? This also speaks to more detail needed as to what exactly was offered to the participants during the ACP conversation.
Reviewer 2 Report
Thank you very much for the opportunity to review this study. I find the subject matter very interesting, however, it requires a number of modifications or clarifications in order to be considered for publication.
Title: The title should be shorter and more informative.
I advise clarifying the type of design with which the study has been carried out, it is not clear whether it is a qualitative study or a clinical trial. Perhaps it should be approached as a qualitative study since the data analysis and results are of this type of study. If so, it is not clear how a qualitative approach or paradigm has not been used (qualitative descriptive study??). This aspect should be clarified. On the other hand, no information is given about the data collection procedure, what were the interviews like, how were they conducted, what were the questions asked of the participants, ......?
What type of sampling has been followed?
In table 1 (description of participants) information should be given for each participant, not for the total number of participants.
Table 2. Is this information really important? Most of the time participants were silent.
Identify in the results the sub-themes.
What implications does this study have for clinical practice?
Reviewer 3 Report
Dear authors.
It was a pleasure to review your manuscript. However, I will point out some issues to improve your paper.
First, the title is too long. I suggest a shorter title.
In the Abstract you should explain where the participants came from, rephrasing sentences. In other words, you must say that participants came from an intervention group of a randomized study and not the inverse.
In data analysis, you explain how transcription and coding were performed. Did you use any coding methodology, for example, content analysis by Bardin, or another author?
You present a difficult subject to work with patients, as they don't understand the concept. In the conclusion, you could suggest a comparison, with other countries/cultures on this topic and analyze why is it so difficult for people to accept ACP.
Best wishes,
Reviewer 4 Report
This is a well written qualitative analiyis of a qualitative data collected druing alreday published as a video-supported nurse-led advance care planning on end-of-life decision-making among frail older patients randomized controlled trial protocol. It should be publsihed as it is.
The manuscript is connected with the previous research done the protocol of which was described and published Leung, D.Y.P.; Chan, H.Y.L.; Yau, S.Z.M.; Chiu, P.K.C.; Tang, F.W.K.; Kwan, J.S.K. A video-supported nurse-led advance 465 care planning on end-of-life decision-making among frail older patients: Protocol for a randomized controlled trial. J Adv 466 Nurs 2019, 75(6), 1360-1369). It consists of thematic analysis of audio-taped nurse-facilitated ACP conversations with frail older adults, and their family members in some cases. The paper is well written and presents the proposed research in an adequate manner. The only clarification needed is that from the authors who should explain in brief under methodology the actual research protocol published under Leung, D.Y.P.; Chan, H.Y.L.; Yau, S.Z.M.; Chiu, P.K.C.; Tang, F.W.K.; Kwan, J.S.K. A video-supported nurse-led advance 465 care planning on end-of-life decision-making among frail older patients: Protocol for a randomized controlled trial. J Adv 466 Nurs 2019, 75(6), 1360-1369)and to explain its connection with this research published in this manuscript? How does the research published in this manuscript fall into the plan of the previously described protocol of the clinical study and whether other results and what has been published so far from this protocol of the clinical study.Author Response
Please see the attachment.

Round 2
Reviewer 2 Report
Thank you very much for allowing me to review this manuscript. The author made the suggested modifications. I consider it a very interesting manuscript.
Author Response
Thanks for the reviewer's appreciation of the revision.